# Monitoring independence in daily life activities after trauma in humanitarian settings: Item reduction and assessment of content validity of the Activity Independence Measure-Trauma (AIM-T)

**Bérangère Gohy**[1,2]*, **Christina H. Opava**[1], **Johan von Schreeb**[3], **Rafael Van den Bergh**[4], **Aude Brus**[5], **Abed El Hamid Qaradaya**[6], **Jean-Marie Mafuko**[7], **Omar Al-Abbasi**[8], **Sophia Cherestal**[9], **Livia Fernandes**[10], **Andre Da Silva Frois**[11], **Eric Weerts**[2], **The AIM-T Study Group**[4,6,7,9,10,11,12,13]¶, **Nina Brodin**[1,14]

1 Division of Physiotherapy, Department of Neurobiology, Care Sciences and Society, Karolinska Institutet, Stockholm, Sweden, 2 Humanity & Inclusion, Rehabilitation technical direction, Brussels, Belgium, 3 Department of Global Public Health, Karolinska Institutet, Stockholm, Sweden, 4 Médecins Sans Frontières, Operational Center Brussels, Brussels, Belgium, 5 Impact & Information Division, Humanity & Inclusion, Innovation, Brussels, Belgium, 6 Médecins Sans Frontières, Operational Center Paris, Gaza, Palestinian territories, 7 Médecins Sans Frontières, Operational Center Brussels, Bujumbura, Burundi, 8 Médecins Sans Frontières, Operational Center Brussels, Erbil, Iraq, 9 Médecins Sans Frontières, Operational Center Brussels, Port-au-Prince, Haiti, 10 Médecins Sans Frontières, Operational Center Paris, Baghdad, Iraq, 11 Médecins Sans Frontières, Operational Center Paris, Paris, France, 12 Médecins Sans Frontières, Operational Center Paris, Aden, Yemen, 13 Humanity & Inclusion, Bujumbura, Burundi, 14 Division of Physiotherapy, Department of Orthopaedics, Danderyd Hospital Corp., Danderyd, Sweden

¶ Membership of the AIM-T study group is provided in the Acknowledgments.
* berangere.gohy@ki.se

## Abstract

A standardized set of measures to assess functioning after trauma in humanitarian settings has been called for. The Activity Independence Measure for Trauma (AIM-T) is a clinician-rated measure of independence in 20 daily activities among patients after trauma. Designed in Afghanistan, it has since been used in other contexts. Before recommending the AIM-T for wider use, its measurement properties required confirmation. This study aims at item reduction followed by content validity assessment of the AIM-T. Using a two-step revision process, first, routinely collected data from 635 patients at five facilities managing patients after trauma in Haiti, Burundi, Yemen, and Iraq were used for item reduction. This was performed by analyzing inter-item redundancy and distribution of the first version of the AIM-T (AIM-T$_1$) item scores, resulting in a shortened version (AIM-T$_2$). Second, content validity of the AIM-T$_2$ was assessed by item content validity indices (I-CVI, 0–1) based on structured interviews with 23 health care professionals and 60 patients in Haiti, Burundi, and Iraq. Through the analyses, nine pairs of redundant items (r$\geq$0.90) were identified in the AIM-T$_1$, leading to the removal of nine items, and resulting in AIM-T$_2$. All remaining items were judged highly relevant, appropriate, clear, feasible and representative by most of participants (I-CVI>0.5). Ten items with I-CVI 0.5–0.85 were revised to improve their cultural relevance or appropriateness and one item was added, resulting in the AIM-T$_3$. In conclusion,

**Data Availability Statement:** The authors confirm that, for approved reasons, some access restrictions apply to the data underlying the findings. Due to the sensitive nature of trauma care (including violent trauma) data, full datasets are not made available by default. Data are available through the MSF Data Sharing Agreement for researchers who meet the criteria for access to confidential data; requests should be addressed to the Data Sharing Agreement coordinator, Annick Antierens (Annick.Antierens@brussels.msf.org).

**Funding:** Enhancing Learning and Research in Humanitarian Action (ELRHA) funded the research coordination (BG) and research activity for this study as part of their Health in Humanitarian Crises (R2HC) call (No.32398), co-funded by the Department for International Development (DFID) and the Wellcome Trust (https://www.elrha.org/programme/research-for-health-in-humanitarian-crises/). The funders had no role in study design, data collection and analysis, decision to publish, or preparation of the manuscript.

**Competing interests:** The authors have declared that no competing interests exist.

the proposed 12-item AIM-T$_3$ is overall relevant, clear, and representative of independence in daily activity after trauma and it includes items appropriate and feasible to be observed by clinicians across different humanitarian settings. While some additional measurement properties remain to be evaluated, the present version already has the potential to serve as a routine measure to assess patients after trauma in humanitarian settings.

## Introduction

Injury is a major cause of burden to society, both in terms of mortality and disability [1]. Injuries have significant negative impact on the socio-economic conditions of the individual, their families, communities, and society. As such, associations with high health expenses, unemployment as well as adversely affected social dynamics have been documented [2–4]. After injury, patients are typically provided trauma care at health facilities. The monitoring and evaluation of quality of trauma care is essential to inform policy and clinical decision making, but also to adapt interventions at different levels [5, 6]. Survival rates, occurrence of secondary complications and 'process indicators', such as length of stay, are commonly used measures to monitor quality of trauma care [7–9]. A patient's functioning, particularly in terms of independence in meaningful daily activities and participation in life situations following trauma care, is now also increasingly recognized [7, 10–13]. However, the most common measures of functioning used in trauma research were not originally developed for patients after trauma [6]. This potentially limits their capacity to fully capture outcomes related to independence in daily activities for patients after trauma [6].

The burden of injury is particularly high in humanitarian contexts, defined as situations in which there is a widespread threat to basic needs exceeding the coping capacity of individuals and communities, including the health care system, due to chronic or sudden-onset crises, caused by natural or technological disasters, famine, epidemics or armed conflict [14]. In such contexts, monitoring is therefore even more essential for appropriate allocation of scant resources [2, 8, 15, 16]. Most validated measures of independence in daily activities have been developed outside of humanitarian contexts and have been reported as impractical in contexts with strained health care system resources, due to high training requirements and long administration time [17]. Additionally, difficulties related to low literacy rates and lack of cultural relevance further hinder the use of existing measures in such contexts [17–20]. Several studies in humanitarian contexts report on functional outcomes using study-specific measures, without description of their measurement properties [21–23].

Previous studies call for a standardized set of measures and associated data collection systems to be used in humanitarian contexts [12, 13, 17, 18, 24]. The inclusion of a short, valid and reliable measure of independence in daily life activities in such a set would allow for time-efficient and improved trauma care monitoring and patient management, development of rehabilitation and surgical protocols, as well as comparative studies between different contexts and joint studies across organizations [17, 18, 25].

A measure of independence in 20 daily life activities, later named the Activity Independence Measure for Trauma (AIM-T), was developed in 2011 by Médecins Sans Frontières (MSF) and Humanity & Inclusion (HI) in the Kunduz Trauma Centre in Afghanistan, a humanitarian context [26]. The AIM-T was initially developed for clinical monitoring in this trauma centre and no formal testing of its validity or reliability was performed. However, the AIM-T was gradually implemented in an increasing number of MSF/HI settings for clinical

and project monitoring purposes. Concerns about its administration time as well as its cultural relevance outside Afghanistan have been raised by health professionals using it routinely.

Our study therefore aimed at reducing the number of items included in the AIM-T, followed by assessment of the content validity of the shortened version, among patients after trauma in different humanitarian contexts.

## Materials and methods

This study consists of two steps: first, item reduction of the original AIM-T (AIM-$T_1$), leading to a shortened version (AIM-$T_2$); second, assessment of content validity of the AIM-$T_2$. The COSMIN methodology was used to document the process [27].

### Activity Independence Measure for Trauma (AIM-T)

The AIM-T was designed in 2011 in the MSF Kunduz Trauma Centre in Afghanistan as a clinician-rated, generic measure of independence in mobility and self-care activities (S1 Fig). It was applied for all patients admitted to the trauma centre: patients of all ages with orthopaedic, visceral or neurological trauma in both inpatient and outpatient care, excluding patients with isolated spinal cord injury or burns [26].

The selection of items for the initial version (AIM-$T_1$) was based on: 1) a review of existing outcome measures [28–31], 2) expert opinion of physiotherapists and, 3) informal patient feedback. Piloting was done in an iterative process and the AIM-$T_1$ included 20 items, divided into two subscales (lower and upper limb). Each item was rated from 1 to 5 with 1 = Total assistance, 2 = Assistance (human support), 3 = Modified independence (use of assistive product), 4 = Independence with difficulties, 5 = Independence. Each subscale score (ranging from 10 to 50) represented the sum of the ten composing items and the total score was the sum of the two subscale scores (ranging from 20 to 100 with higher scores indicating higher independence).

### Study setting

The following study centres located in different humanitarian contexts were selected for the present study.

- The MSF trauma centres of Arche (Burundi), Tabarre (Haiti) and Aden (Yemen) set up according to the MSF trauma centre model [32, 33], where physiotherapy is provided in both in- and outpatient care.

- The MSF Baghdad Medical Rehabilitation Centre (Iraq), MSF Mosul Comprehensive Post-operative Care Centre, in collaboration with HI (Iraq), and the MSF-HI Haguruka Rehabilitation Centre (Burundi), providing post-operative care. Physiotherapy was provided in all three centres to both in- and outpatients.

### Study population

For item reduction, the six centres routinely using the AIM-$T_1$ (Arche, Tabarre, Aden, Baghdad and Haguruka) were included. All patients receiving physiotherapy as part of their trauma care and having at least one AIM-T score from one or several of the four timepoints (i.e., physiotherapy inpatient department (IPD) admission, physiotherapy IPD discharge, physiotherapy outpatient department (OPD) admission, physiotherapy OPD discharge) collected between August 2017 and July 2018 were included. Orthopaedic, visceral and soft tissue injuries were categorised as injuries of pelvis and lower limb, upper limb and trunk (i.e., spine, abdomen, and chest).

For content validity assessment, patients and healthcare professionals (HCP) from four centres (Tabarre, Arche, Baghdad and Mosul) were included between March and June 2019. The centres were selected for diversity in terms of culture and geographic region and data collection feasibility. Participant recruitment was based on purposive sampling to ensure diversity of characteristics. All included patients had sustained orthopaedic, visceral and/or soft tissue injuries, were receiving care at one of the included centres and had a minimum age of five years. Diversity was sought in terms of age, gender, injury location, nature, severity, acuteness, and triage colour using the South African Triage Score classifying the severity of injury from 'green' i.e. minor injuries, to 'red' i.e. emergency to be seen immediately [34], with a target of 15 patients per centre. All included HCPs worked in the study centres as either physiotherapist, nurse, or medical doctor. Diversity was sought in terms of age, gender, years of experience in trauma care, knowledge about the AIM-$T_1$ and area of specialization, aiming at five HCPs per centre.

All participants gave their informed consent before starting the interviews, both in terms of participation and interview recording. For patients under 18 and/or with cognitive difficulties the informed consent of the patient's representative was requested. Additionally, an assent form was requested from 12- to 18-year-old patients. For patients with literacy difficulties and/or wanting to participate but refusing to give written consent, a witness was used to confirm verbal consent. If a participant did not consent to recording, notes were taken during the interview by the interviewer and a second trained person.

## Data collection and variables

Item reduction was performed on data collected routinely in the six study centres. These data included the 20 items of the AIM-$T_1$ at a minimum of one of the four time points, socio-demographic information, and routine clinical data on the injuries. Data were available in paper registers and encoded in a dedicated study database. The final study dataset consisted of pooled data across all centres and across time points at which the AIM-$T_1$ was recorded.

Content validity assessment was performed at the four selected study centres through individual, structured interviews. The interviews were conducted by one interviewer per centre who was medically literate, spoke the local language and was trained by the first author regarding interview technique and use of the interview grid. The interview grids assessed the extent to which the AIM-$T_2$ comprehensively and adequately measured independence in daily life activities among patients after trauma, as per the COSMIN methodology for assessing content validity [35]. To gather spontaneous insights from the participants on the construct (independence in daily activities), patients were first asked to rate their level of independence on a visual analogue scale (VAS, 0 = totally dependent, 100 = totally independent) without having seen the AIM-$T_2$ and then to explain their rating by 'thinking aloud'. Similarly, HCPs were asked to rate and elaborate on the level of independence of three of their patients, still not referring specifically to the AIM-T. Participants were then shown the AIM-$T_2$ and asked to rate each item on a 1 ('not at all') to 4 ('highly') scale regarding two components: the activity's *relevance* in daily life as well as the *appropriateness* of observing/being observed in the activity (only participants above 18). HCPs were asked to use the same scale to rate each item for three additional components: its *clarity*, *feasibility* to observe, and *representativeness* in reflecting independence in upper and lower limb activities respectively, as well as the clarity of the existing AIM-$T_2$ scoring system. All participants were invited to comment on each of their ratings. Each interview was concluded with general questions on the comprehensiveness of each subscale, whether any items should be deleted or added, and any other comments were encouraged.

Interviews were performed in the participants' native languages, recorded and transcribed verbatim. At the same time, they were translated into English for Iraq, and into French for Burundi and Haiti. Interview notes from those not consenting to be recorded were compiled, translated and encoded verbatim.

## Data analyses

Descriptive data was presented using frequencies, medians with interquartile ranges (IQR), or means and standard deviations (SD) depending on type of data. Correlations were analyzed with the Spearman rank correlation coefficients. All quantitative analyses were performed in SPSS version 27.

For item reduction, *inter-item redundancy* and *distribution* of the AIM-T$_1$ item scores were investigated. *Inter-item redundancy* was analysed by correlations of five pairs of lower limb items, hypothesized to be redundant since assessing similar constructs, assessed among patients with at least one lower limb or trunk injury. Similarly, five pairs of upper limb items hypothesised to be redundant were assessed among patients with at least one upper limb or trunk injury [36]. Pairs of items with a correlation coefficient equal to or greater than 0.9 were considered redundant. The *distribution of item scores* was assessed through the proportions of minimum (floor effect) and maximum scores (ceiling effect) of single items at inpatient admissions and outpatient discharges. Most patients with trauma can be expected to be less independent in activities (lower AIM-T scores) at admission and more independent (higher scores) at discharge. Thus, to make the AIM-T better suited to evaluate changes of independence in activities along the continuum of care, the item within highly redundant pairs with a larger ceiling effect at inpatient admission and/or floor effect at outpatient discharge was deleted from the next AIM-T version [37]. In case of similar item distribution, the two redundant items were merged. *Refinement of wording* and adjustment to clinical knowledge of some items was considered after consultation with clinical teams.

For content validity of the AIM-T$_2$, units of information were extracted by a research assistant from the transcribed answers to the introductory interview question. The WHO International Classification of Functioning, Health and Disability (ICF) defines "Functioning in the context of health, as an umbrella term for body functions, body structures, activities and participation", and describes the activities as "the tasks and actions executed" in different life situations. Each unit of information was related to ICF codes, with a focus on the Mobility and Self-care Activity domains. To ensure objectivity, the ICF coding was performed independently by the research assistant and the first author and negotiated consent was used to resolve any discrepancy. Concepts within the targeted ICF domains that were suggested, either in the introductory question or in the question on which items to add, by more than 15% of participants were considered for addition to a revised AIM-T$_3$ [38]. Aspects of feasibility and appropriateness of observing the suggested activities were also considered when revising the AIM-T$_2$ items.

A quantitative analysis of the participants' ratings of each AIM-T$_2$ item in relation to the five components (i.e. relevance, appropriateness, clarity, feasibility and representativeness) was performed using *item content validity indexes* (I-CVI) [39]. The I-CVI is calculated as the number of participants scoring 4 for an item in relation to each of the components, divided by the total number of participants scoring the item; resulting in five I-CVIs between 0 and 1 for each item. Items with any I-CVI lower than 0.5 were considered for removal from the revised AIM-T$_3$ and items with any I-CVI between 0.5 and 0.85 were considered for revision in relation to that specific component. Comments by the participants when rating items were used to guide the revision of the items and the ICF terminology guided the reformulation of items.

### Ethics

The protocol of this study was approved by the MSF Ethics Review Board, Geneva, Switzerland (ID 1893) and the Swedish Ethical Review Authority (Dnr 2022-02806-01), as well as by the respective ethics review committees competent for each participating centre: University of Aden (REC-53-2019), Ethics Committees of the Baghdad Directorate of Health and the Ninewa Directorate of Health (p. 1/5/10), Burundi National Ethics Committee for the Protection of Human Rights of Participants in Biomedical (15/04/2019) and Behavioral Research and Haiti National Committee of Bioethics (Ref.1819-23).

## Results

### Participants

The item reduction included 635 patients after trauma (median age 25 years, IQR 16–35 years, 17.2% female). Lower limbs were most often affected (59%), in isolation or combination with other injuries, with single lower limb fractures being most frequent (40%). Violent trauma was the most common cause of injury (41%), with the vast majority due to gunshot injuries (28%), followed by road traffic accidents (31%). More information on the patients' characteristics can be found in Table 1. For these patients, 1207 AIM-$T_1$ scores were collected: 463 from Aden,

**Table 1. Characteristics of the 635 patients included in the item reduction analysis.**

| Characteristics | Patient N (%) |
|---|---|
| **Centre (country)** | |
| Aden (Yemen) | 231 (36.4) |
| Baghdad (Iraq) | 145 (22.8) |
| Tabarre (Haiti) | 73 (11.5) |
| Arche and Haguruka (Burundi) | 186 (29.3) |
| **Age (median IQR)** | 25.0 (16–35) |
| <17 | 166 (26.1) |
| 18–45 | 367 (57.8) |
| >45 | 73 (11.5) |
| Missing | 29 (4.6) |
| **Sex** | |
| Male | 525 (82.6) |
| Female | 109 (17.2) |
| Missing | 1 (0.2) |
| **Trauma location**[a] | |
| ≥ 1 lower limb injury | 373 (58.7) |
| ≥ 1 upper limb injury | 231 (36.4) |
| ≥ 1 trunk injury | 83 (13.1) |
| **Cause of injury** | |
| Road traffic accident | 199 (31.3) |
| Fall | 123 (19.4) |
| Fire | 3 (0.5) |
| Gunshot | 176 (27.7) |
| Bomb/mine | 56 (8.8) |
| Knife | 19 (3.0) |
| Assault/torture | 10 (1.6) |
| Other | 36 (5.7) |
| Missing | 13 (2.0) |

[a]Patients could have injuries in more than one area

227 from Baghdad, 227 from Tabarre and 290 from Arche and Haguruka, across the four time-points: 408 IPD admission (34%), 272 IPD discharge (22%), 312 OPD admission (26%) and 215 OPD discharge (18%).

For content validity, 83 participants were interviewed. Their characteristics are shown in Table 2.

**Table 2. Characteristics of interview participants for content validity; 60 patients and 23 health care professionals (HCP).**

| Characteristics | Patients N (%) | HCP N (%) |
|---|---|---|
| **Centre (country)** | | |
| Mosul (Iraq) | 10 (16.7) | 5 (21.7) |
| Baghdad (Iraq) | 15 (25.0) | 8 (34.9) |
| Arche (Burundi) | 21 (35.0) | 5 (21.7) |
| Tabarre (Haiti) | 14 (23.3) | 5 (21.7) |
| **Age (median IQR)** | 28.5 (18.75–40.25) | 31 (28–36) |
| 5–17 | 13 (21.7) | 0 |
| 18–45 | 39 (65.0) | 22 (95.7) |
| >45 | 6 (10.0) | 1 (4.3) |
| Missing | 2 (3.3) | 0 |
| **Sex** | | |
| Male | 41 (68.3) | 15 (65.2) |
| Female | 19 (31.7) | 8 (34.8) |
| **Trauma location[a]** | | N.A. |
| ≥ 1 lower limb injury | 44 (73.3) | |
| ≥ 1 upper limb injury | 21 (35.0) | |
| ≥ 1 trunk injury | 9 (15.0) | |
| Missing | 5 (8.3) | |
| **Trauma acuteness** | | N.A. |
| < 30 days | 11 (18.3) | |
| 30 days—6 months | 13 (21.7) | |
| >6 months | 28 (46.7) | |
| Missing | 8 (13.3) | |
| **SATS[b] Triage color** | | N.A. |
| Green | 2 (3.3) | |
| Yellow | 15 (25.0) | |
| Orange | 14 (23.4) | |
| Red | 9 (15.0) | |
| Missing | 20 (33.3) | |
| **Profession** | N.A. | |
| Physiotherapist | | 14 (60.9) |
| Medical doctor | | 5 (21.7) |
| Nurse | | 4 (17.4) |
| **Self-reported knowledge and use of AIM-T** | N.A. | |
| None | | 8 (34.8) |
| Basic | | 4 (17.4) |
| Good | | 10 (43.5) |
| Advanced | | 1 (4.3) |

[a]Patients could have injuries in more than one area
[b]South African Triage Score (SATS), N.A. = not applicable

## Item reduction

Within the ten pairs of AIM-$T_1$ items analysed for hypothesized inter-item redundancy, one pair was not redundant and both items were therefore kept. Out of the nine remaining pairs, one item was removed from each of seven pairs and items from two pairs were merged, based on the distribution of item scores (Table 3). Some items and scoring levels were reformulated based on the ICF terminology and after field team feedback for clarity, to be more gender inclusive or better adapted to cultural context, leading to the AIM-$T_2$.

## Content validity

Fifty-five patients rated their level of independence as median 53.0 (IQR = 33–73) and HCP rated the independence of 68 patients as median 70.0 (IQR = 48.5–80). In the subsequent 'think-aloud' exercises, 1074 units of information were identified. After deleting duplications within each single interview, the 670 remaining units were related to ICF codes, among which 306 related to the domains of Mobility or Self-care. Environmental factors of importance for daily activities, i.e. caregivers and/or assistive products, were mentioned within 55% and 58% of the patient and HCPs think-aloud descriptions respectively, as influencing the perceived level of independence. Difficulties in performing activities such as pain, time needed, quality of movement, safety, or smoothness was mentioned by 39% of participants. The only two activities not already included in the AIM-$T_2$ that were mentioned by >15% of the study

**Table 3. Spearman correlation coefficients ($r^s$) for selected pairs of AIM-$T_1$ items within the lower limb and upper limb subscales in 635 patients (1207 assessments) and proportions of maximum AIM-$T_1$ scores at admission to inpatient departments (IPD) for lower (n = 321) and upper (n = 186) limb item scores and of minimum scores at discharge from outpatient departments (OPD) for lower (n = 91) and upper (n = 135) limb item scores.**

| | $r^s$ | Maximum item scores at IPD admission N (%) | Minimum item scores at OPD discharge N (%) | Revisions for AIM-$T_2$ |
|---|---|---|---|---|
| **Lower limb subscale** | | | | |
| **Walk around <50 m** | 0.94 | 17 (5.3) | 0 (0) | Walk 50 metres |
| **Walk around >50 m** | | 18 (5.6) | (0) | |
| **Go up stairs** | 0.99 | 15 (4.7) | 0 (0) | Go up and down 5 steps |
| **Go down stairs** | | 15 (4.7) | 0 (0) | |
| **Lie down** | 0.89 | 71 (22.1) | 0 (0) | Lie down |
| **Sit up** | | 59 (18.4) | 0 (0) | Sit up |
| **Stand up** | 0.93 | 30 (9.3) | 0 (0) | Stand up |
| **Sit down** | | 40 (12.5) | 0 (0) | |
| **Full squat** | 0.94 | 27 (8.4) | 4 (4.4) | Full squat |
| **Kneel** | | 27 (8.4) | 5 (5.5) | |
| **Upper limb subscale** | | | | |
| **Open jar** | 0.92 | 52 (28.0) | 0 (0) | Open jar |
| **Grab cup** | | 55 (29.6) | 0 (0) | |
| **Grab pen** | 0.95 | 71 (38.2) | 0 (0) | Grab small object |
| **Thumb opposition** | | 72 (38.7) | 0 (0) | |
| **Eat** | 0.90 | 51 (27.4) | 0 (0) | Lifting and carrying object above shoulder level |
| **Carry overhead** | | 44 (23.7) | 0 (0) | |
| **Wash back** | 0.90 | 41 (22.0) | 1 (0.7) | Grooming |
| **Comb hair** | | 45 (24.2) | 1 (0.7) | |
| **Put on pants** | 0.93 | 41 (22.0) | 0 (0) | Put on pants |
| **Put on shirt** | | 41 (22.0) | 0 (0) | |

participants were toileting (22%) and walking long distances (18%). These two activities were therefore considered for addition to the AIM-T$_3$.

All AIM-T$_2$ items were rated as highly relevant, appropriate, clear, feasible and representative by the majority of the participants (Table 4). However, ten AIM-T$_2$ items had I-CVIs lower than 0.85 for one or more components and were considered for revision in accordance with comments and to fit with ICF terminology (Table 4). In regards to comprehensiveness of subscales, 36 (43%) and 37 (45%) participants identified the need to add activities in the lower and upper limb subscales, respectively. However, there was no activity suggested by more than 15% of participants. Only a minority of participants thought that any activity should be removed from the AIM-T. Considering feasibility and appropriateness, 'toileting' was not included in AIM-T$_3$ while 'timed 10 metre walk/move around' was added, as a proxy for 'walking long distances'. Table 4 presents the revisions of items based on the I-CVIs and grouping of core items into a third subscale (i.e. "core subscale"). Based on participants' description of difficulties in relation to their independence in the introductory question and on their feedbacks on the AIM-T$_2$, the scoring level 3 'modified independence (use of assistive product)' was split into two scoring levels. The scoring system now includes six levels (0 = totally dependent, 1 = dependent on human support, 2 = dependent on equipment/environment modification with difficulties, 3 = dependent on equipment/environment modification without difficulties, 4 = independent with difficulties, 5 = totally independent). The revised AIM-T$_3$ is composed of 12 items grouped into three subscales, and ranges from 0 to 60 is found in S2 Fig.

## Discussion

This study aimed at item reduction and content validity assessment of the AIM-T in patients after trauma in different humanitarian contexts. High correlations between nine out of ten pairs of items were found in the first version of the AIM-T. This finding was used to shorten and revise the measure accordingly. This study showed that the revised version of the AIM-T is relevant, clear and representative of independence in daily activities after trauma, supporting its content validity across a range of humanitarian contexts. Having a shorter measure is crucial for settings with time constraints and limited resources, while revisions elicited by content validity assessment enabled items that are appropriate and feasible to be observed by clinicians across different humanitarian contexts.

High ceiling effects were observed among the AIM-T$_1$ upper limb activities. The AIM-T was designed to be used across the continuum of care, requiring some items to be more sensitive in the acute stage, and others in the post-acute stages, inevitably leading to floor and ceiling effects [37, 40, 41]. Moreover, ceiling effects has been described previously in other generic upper limb measures, with the challenge of finding more difficult upper limb items that are relevant to a heterogeneous population [42, 43]. This was considered when revising the AIM-T activities, using information shared by participants during the interviews, and it should remain a point of focus in future studies.

Validated outcome measures commonly used in the field of rehabilitation and developed outside humanitarian contexts, such as the Functional Independence Measure or the Barthel Index, often contain self-care activities. The interviews however indicated that self-care activities 'Put on pants', 'Grooming' and 'Toileting' were relevant for patients in their daily life, though not always culturally appropriate for observation. Moreover, these self-care activities are performed differently across genders, settings and cultures [20, 44, 45]. Identifying the specific movements necessary to perform those activities in such contexts is therefore crucial to find appropriate activity substitutes, such as 'reaching lower back and grasp clothes' instead of 'put on pants' [46]. The AIM-T$_1$ self-care activities were consequently replaced by mobility

**Table 4. Item Content Validity Indices (I-CVI) of the AIM-T$_2$ items in terms of relevance, appropriateness, clarity, feasibility and representativeness, as expressed by patients (n = 60) and healthcare professionals (n = 23).** Comments on I-CVIs <0.85 (in bold) are included as well as revisions of items for AIM-T$_3$.

| AIM-T$_2$ items | Relevance N = 83 | Appropriateness N = 70[1] | Clarity N = 23 | Feasibility N = 23 | Representativeness N = 23 | Comments on Relevance (R), Appropriateness (A), Clarity (C), Feasibility (F) or Representativeness (RE) | AIM-T$_3$ items |
|---|---|---|---|---|---|---|---|
| | | | | | Lower limb subscale | | |
| **Walk 50 m** | **0.81** | 0.97 | 0.87 | 0.87 | 0.83 | **R**: Relevant for indoor walking, but not for longer distances required for daily community ambulation | Walk/move around 14 metres Timed 10 metre walk/move around |
| **Go up and down 5 steps** | **0.59** | 0.93 | 0.86 | **0.83** | **0.65** | **R**: No stairs in home and rural environments, but hills and obstacles **F**: Challenging to observe and handle patients' fears in hospital **RE**: Need to increase number of steps | Climb up and down 10 steps |
| **Sit up** | 0.85 | 0.93 | **0.83** | 0.91 | 0.91 | **C**: Precisions needed as to use of arms and support. Not specific to lower limb function | Sit up and remain seated for 10 seconds[2] |
| **Stand up** | 0.96 | 0.96 | 0.91 | 1 | 1 | | Stand up and remain standing for 10 seconds |
| **Lie down** | **0.83** | 0.93 | 0.91 | 1 | 0.91 | **R**: Associated with sleeping and non-active transfer. Not specific to lower limb function | Roll over[2] |
| **Squat** | **0.73** | **0.51** | 0.87 | 0.87 | **0.57** | **R and RE**: Not performed daily across contexts **A**: Culturally inappropriate to be observed in that position, especially for women, as squatting is often associated to toileting position | Kneel down and stand up |
| | | | | | Upper limb subscale | | |
| **Open jar** | **0.68** | 0.98 | 0.96 | 1 | **0.83** | **R and RE**: Not performed daily, while twisting open different types of objects, such as a bottle, is | Open a jar/bottle |
| **Grab small object** | **0.79** | 1 | 0.87 | 1 | 1 | **R**: Specific to hands and not overall upper limb. Some need to handle bigger objects. Precision of size and weight of object needed | Pick up small object and manipulate) |
| **Lift and carry object above shoulder level** | **0.72** | 0.96 | 0.91 | 0.91 | **0.78** | **R and RE**: Not performed daily across patients, but needed when putting on shelf, fetching water, placing on head and for construction work. Precision of shape and weight of object required | Lift and carry 5 kg above shoulder level |
| **Groom** | 0.96 | **0.84** | 0.87 | 0.87 | 0.96 | **A**: Related to private sphere, especially for women | Reach face and neck |
| **Put on pants** | **0.77** | **0.55** | 0.91 | **0.74** | 0.91 | **R**: Not applicable to all, especially for women and when injured. Rather a lower limb activity **A**: Related to private sphere, embarrassing for women to be observed by opposite gender **F**: For the above reasons | Reach lower back and grasp clothes |

[1] Children (n = 13) did not answer this question

[2] Items moved to a third subscale 'Core subscale'

activities in the AIM-$T_2$, to address cultural sensitivity, feasibility as well as to enhance comparability across cultures.

Participants frequently mentioned that community ambulation involving walking long distances was important for participation in many life situations, such as going to school, fetching water, or attending religious ceremonies, activities or events. However, this cannot be easily observed in a clinical setting. The activity 'Timed 10-metre walk/move around' was added to the AIM-$T_3$ as proxy for walking long distances, using 12.5 seconds as a cut-off value for higher independence. Indeed, this walking speed has been previously used as indicator for community ambulation ability [47, 48]. 'Walking short distances' was also reported as relevant, and striving for simplicity, the AIM-$T_2$ item 'Walking 50 metres' was modified to 'Walk/Move around 14 metres' in the AIM-$T_3$. This allowed the two walking activities to be scored simultaneously by timing the middle 10 metres of a 14-metre walk, as per the 10 metre walk test protocol used by other authors, while also observing the level of assistance required for the entire walk [47].

Our study participants also expressed how the use of assistive products or need for human assistance influenced their perception of daily life independence. This is even more critical in humanitarian settings, where loss of available support, e.g. loss of assistive product or displacement of relatives, makes individuals more vulnerable [49]. One important advantage of AIM-T over existing measures such as the Barthel Index and the World Health Organization Disability Assessment Schedule 2.0 (WHODAS 2.0) is that use of assistive products are taken into consideration in the scoring system [30, 50].

This study has several strengths. Our study populations include large samples of patients from a diversity of humanitarian contexts, increasing the applicability of the findings to other humanitarian contexts [13, 51]. Moreover, our sample was predominantly composed of young males with extremity injuries, which mirrors the population with trauma in those settings [52]. The use of COSMIN methodology has provided us with a recognised framework to assess measurement properties in a structured way. More specifically, the assessment of content validity of the AIM-$T_2$ early on in the process is a strong asset as it is an important prerequisite for reliability, internal consistency and interpretability [35]. Gathering input from both expert HCPs, as well as from patients with activity limitations following trauma has improved the relevance and acceptance of the measure. Interviews were conducted by one local trained interviewer per centre. Although this may have led to personal differences in interview styles, it is certainly a strength of our study since it ensured that all interviews were conducted with a full understanding of the local beliefs, values and language, and heterogeneity of results was minimised by using the same interview grid and training package for all interviewers [53, 54].

This study was not without limitations. While we generally adhered to COSMIN methodology, resource and time constraints at the operational level precluded some requirements such as the re-evaluation of measurement properties established in a previous step, as recommended by COSMIN [41]. However, the content validity was already supported by I-CVI > 0.50 before the last revision and could be assumed to have been further improved since the revision of items was based on both patients' and HCPs' comments. The interviews were only conducted in three humanitarian contexts, which, in a strict sense, means that content validity cannot be confirmed beyond those. On the other hand, the diversity of contexts included may increase the external validity of our findings. Translations of the interview transcriptions from local languages to English or French might have introduced bias [53]. However, the persons in charge of the translation were medically literate, with a good knowledge of the population studied and the content of the interviews was very concrete and short, limiting the room for interpretation. Finally, routine data, collected by HCP who were trained locally to use the AIM-T, was used for item reduction. While this has the advantage of representing

data collection in daily clinical practice, it may also have influenced the quality of data. This has highlighted the need for a more structured training package, which has since been developed to limit potential misunderstandings in future data collection.

With many measures assessing independence in daily activities already existing, the relevance of introducing a new measure could be challenged [55]. However, we suggest that the AIM-T addresses a previously unmet need and combines a set of unique features. As a generic measure, it reduces the need for extensive training on a multitude of measures. Additionally, it is open access and free of licensing, which enhances its use in similar contexts. This clinician-rated measure includes activities that are culturally appropriate and relevant across different humanitarian settings as well as feasible to be observed in clinical settings. The content validity testing of the AIM-T was a crucial step but further investigation of its validity and reliability remains essential before it can be recommended for wider use. Nevertheless, the present version already has the potential to serve as a routine measure to assess patients after trauma in humanitarian contexts for clinical decision making and patient evaluation, and we therefore recommend its use.

## Supporting information

**S1 Fig. Activity Independence Measure–Trauma, first version (AIM-T$_1$).** (DOCX)

**S2 Fig. Activity Independence Measure–Trauma, third version (AIM-T$_3$).** (DOCX)

## Acknowledgments

We would like to thank the local teams in each of the centres, as well as the coordination and headquarters teams at both Médecins Sans Frontières and Humanity & Inclusion, who despite working in challenging environments have provided their commitment and support to make this study happen. More particularly, we would like to thank the interviewers as well as Audrey Desclee, Gaëlle Smith, Dr Mohammed Abu Mughiaseeb, Albert Nikiema, Amber Alayyan, Ayokunnu Raji, Khaled Ahmedana, Vincent Lambert, Gbane Mahama, and Sandrina Simons for their valuable contributions to this research project. We would also like to thank our study participants who took their time to participate in the content validity interviews. This study has been funded by Elrha's Research for Health in Humanitarian Crises (R2HC) Programme, which aims to improve health outcomes by strengthening the evidence base for public health interventions in humanitarian crises. R2HC is funded by the UK Foreign, Commonwealth and Development Office (FCDO), Wellcome, and the UK National Institute for Health Research (NIHR).

The following co-authors are part of the AIM-T study group: Nathalie Hanssens, Evelyne Côté Grenier, Eric Ndiramiye, Sylvia Sommella, Minolta Florville, Valeria Maglia, Julie Van Hulse, Annick Antierens and Clair Mills.

## Author Contributions

**Conceptualization:** Bérangère Gohy, Christina H. Opava, Johan von Schreeb, Rafael Van den Bergh, Aude Brus, Andre Da Silva Frois, Eric Weerts, Nina Brodin.

**Data curation:** Bérangère Gohy.

**Formal analysis:** Bérangère Gohy, Nina Brodin.

**Funding acquisition:** Bérangère Gohy, Christina H. Opava, Johan von Schreeb, Rafael Van den Bergh, Aude Brus, Andre Da Silva Frois, Eric Weerts, Nina Brodin.

**Investigation:** Bérangère Gohy, Abed El Hamid Qaradaya, Livia Fernandes, Eric Weerts.

**Methodology:** Bérangère Gohy, Christina H. Opava, Johan von Schreeb, Rafael Van den Bergh, Aude Brus, Andre Da Silva Frois, Eric Weerts, Nina Brodin.

**Project administration:** Bérangère Gohy, Rafael Van den Bergh, Abed El Hamid Qaradaya, Jean-Marie Mafuko, Omar Al-Abbasi, Sophia Cherestal, Andre Da Silva Frois, Eric Weerts, Nina Brodin.

**Supervision:** Bérangère Gohy, Jean-Marie Mafuko, Omar Al-Abbasi, Sophia Cherestal, Andre Da Silva Frois, Eric Weerts, Nina Brodin.

**Validation:** Bérangère Gohy, Christina H. Opava, Johan von Schreeb, Rafael Van den Bergh, Aude Brus, Andre Da Silva Frois, Eric Weerts, Nina Brodin.

**Writing – original draft:** Bérangère Gohy, Christina H. Opava, Johan von Schreeb, Rafael Van den Bergh, Aude Brus, Nina Brodin.

**Writing – review & editing:** Bérangère Gohy, Christina H. Opava, Johan von Schreeb, Rafael Van den Bergh, Aude Brus, Abed El Hamid Qaradaya, Jean-Marie Mafuko, Omar Al-Abbasi, Sophia Cherestal, Livia Fernandes, Andre Da Silva Frois, Eric Weerts, Nina Brodin.

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
