## [Decision Letter · Decision Letter 0]

14 Oct 2022

PGPH-D-22-01425

Monitoring independence in daily life activities after trauma in humanitarian settings: Item reduction and assessment of content validity of the Activity Independence Measure-Trauma (AIM-T)

Dear authors

Thank you for submitting your manuscript to PLOS Global Public Health. After careful consideration, we feel that it has merit but does not fully meet PLOS Global Public Health’s publication criteria as it currently stands. Therefore, we invite you to submit a revised version of the manuscript that addresses the points raised during the review process.

We look forward to receiving your revised manuscript.

Kind regards,

Andreas K Demetriades, MBBChir, MPhil, FRCSEd, FEBNS.

Academic Editor

Journal Requirements:

2. Please send a completed 'Competing Interests' statement, including any COIs declared by your co-authors. If you have no competing interests to declare, please state "The authors have declared that no competing interests exist". Otherwise please declare all competing interests beginning with the statement "I have read the journal's policy and the authors of this manuscript have the following competing interests:"

3. We do not publish any copyright or trademark symbols that usually accompany proprietary names, eg ©, ®, or ™  (e.g. next to drug or reagent names). Please remove all instances of trademark/copyright symbols throughout the text, including ® on page 33.

Additional Editor Comments (if provided):

The peer review has made some recommendations which are constructive

Please revise and resubmit

Reviewers' comments:

Reviewer's Responses to Questions

**Comments to the Author**

1. Does this manuscript meet PLOS Global Public Health’s publication criteria? Is the manuscript technically sound, and do the data support the conclusions? The manuscript must describe methodologically and ethically rigorous research with conclusions that are appropriately drawn based on the data presented.

Reviewer #1: Yes

Reviewer #2: Yes

2. Has the statistical analysis been performed appropriately and rigorously?

Reviewer #1: Yes

Reviewer #2: Yes

3. Have the authors made all data underlying the findings in their manuscript fully available (please refer to the Data Availability Statement at the start of the manuscript PDF file)?

Reviewer #1: Yes

Reviewer #2: No

4. Is the manuscript presented in an intelligible fashion and written in standard English?

Reviewer #1: Yes

Reviewer #2: Yes

5. Review Comments to the Author

Reviewer #1: I am of the opinion from what is presented in the manuscript in all sections, that the authors have presented a well written report of their work. The abstract is clear, the introduction and methodological approach appears good.

Reviewer #2: In my view, the study described in the manuscript addresses a pragmatic and relevant topic, which is the development and validation of a tool to uniformly measure functioning after trauma in humanitarian contexts. A plus is an attention the authors had in trying to overwhelm some limitations of the existing measures (i.e. some activities may be present in some cultures while not in others) and to propose a tool that might be applied by the different stakeholders in humanitarian contexts.

Overall, the study consisted of two steps: (i) reduction of the number of items of the original version of a previously designed scoring system (AIM-T1); (ii) assessment of the content validity of the new scoring system with reduced items (AIM-T2) and development of a new, validated tool (AIM -T3).

During my critical appraisal, I made the following observations

Abstract

The abstract clearly introduces the topic and the objectives of the study and well summarises the adopted methodology as well as the main results.

Introduction

The introduction is overall well written and logically defines the problem that the study aims to solve.

Materials and methods:

The methodology is described in detail and follows a clear logical order of the item reduction and content validity assessment steps. The authors are aware that some requirements of the COSMIN methodology for assessing content validity were not followed, but clearly state this in the limitations (lines 374-377) and motivate why the content validity can be anyway supported (lines 377-379).

Just some minor comments:

• Is some more information about the location (i.e. head injury, pelvic injury, thoracic injury…), severity (Injury Severity Score?) and mechanism (i.e., violence, car accident..) of the injury available? If yes, a table could be added referring to lines 248-255 and also added to the groups in table 1.

• Lines 135-137: how many AIM-T scores were available for IPD admission and discharge and OPD admission and discharge?

• Line 223: it’s not clear to me why a 15% threshold was chosen for the addition of an item: are there references about this percentage?

Results:

Just some minor comments:

• Lines 284-287: why toileting and walking long distances, although mentioned by >15% of the participants were not included in the final version of the questionnaire? Also, I don’t see them in table 3.

• Lines 298-303: why the scoring system was expanded by one level (5�6)? I read “based on participant’s description…and feedback..”, but what does this mean?

• Some of the AIM-T3 items reported in the last column of table 3 are slightly different from those reported in the final version of the questionnaire (i.e. sit up and remain seated for 10 seconds vs sit up (supine edges); stand up and remain standing for 10 seconds vs stand up (sit to stand); lift and carry 5 kg above shoulder level vs lift 5 kb above shoulder level…). Which is the correct version?

• How was the 14 meters distance chosen? Is this the correspondent of walking long distances or is it something in between short and long distances?

Discussion:

The discussion is well-written and clearly summarises the relevance of the AIM-T3, as well as its limits and pros. The authors are aware that this tool will need further validation for wider use in different scenarios and clearly state this in the discussion.

6. PLOS authors have the option to publish the peer review history of their article (what does this mean?). If published, this will include your full peer review and any attached files.

**Do you want your identity to be public for this peer review?** For information about this choice, including consent withdrawal, please see our Privacy Policy.

Reviewer #1: **Yes: **Nnodimele Onuigbo ATULOMAH, PhD

Reviewer #2: No

---

## [Decision Letter · Decision Letter 1]

22 Nov 2022

Monitoring independence in daily life activities after trauma in humanitarian settings: Item reduction and assessment of content validity of the Activity Independence Measure-Trauma (AIM-T)

PGPH-D-22-01425R1

Dear Gohy,

We are pleased to inform you that your manuscript 'Monitoring independence in daily life activities after trauma in humanitarian settings: Item reduction and assessment of content validity of the Activity Independence Measure-Trauma (AIM-T)' has been provisionally accepted for publication in PLOS Global Public Health.

Best regards,

Andreas K Demetriades, MBBChir, MPhil, FRCSEd, FEBNS.

Academic Editor

Thank you for your submission and for addressing the points raised by the peer review process

I am happy to recommend this manuscript for publication

Reviewer Comments (if any, and for reference):

Reviewer's Responses to Questions

**Comments to the Author**

1. If the authors have adequately addressed your comments raised in a previous round of review and you feel that this manuscript is now acceptable for publication, you may indicate that here to bypass the “Comments to the Author” section, enter your conflict of interest statement in the “Confidential to Editor” section, and submit your "Accept" recommendation.

Reviewer #2: All comments have been addressed

2. Does this manuscript meet PLOS Global Public Health’s publication criteria? Is the manuscript technically sound, and do the data support the conclusions? The manuscript must describe methodologically and ethically rigorous research with conclusions that are appropriately drawn based on the data presented.

Reviewer #2: Yes

3. Has the statistical analysis been performed appropriately and rigorously?

Reviewer #2: Yes

4. Have the authors made all data underlying the findings in their manuscript fully available (please refer to the Data Availability Statement at the start of the manuscript PDF file)?

Reviewer #2: Yes

5. Is the manuscript presented in an intelligible fashion and written in standard English?

Reviewer #2: Yes

6. Review Comments to the Author

Reviewer #2: Dear Authors,

Thank you for having addressed all my comments and suggestions in the manuscript and for having clarified some parts of the text.

7. PLOS authors have the option to publish the peer review history of their article (what does this mean?). If published, this will include your full peer review and any attached files.

**Do you want your identity to be public for this peer review?** For information about this choice, including consent withdrawal, please see our Privacy Policy.

Reviewer #2: **Yes: **Nicoló Marchesini
